# How **NOT** to benchmark your SITE metric: Beyond Static Leaderboards and Towards Realistic Evaluation.

**Prabhant Singh**
*p.singh@tue.nl*
AMOR/e Lab
TU Eindhoven

**Sibylle Hess**
*s.c.hess@tue.nl*
TU Eindhoven

**Joaquin Vanschoren**
*j.vanschoren@tue.nl*
AMOR/e Lab
TU Eindhoven

## ABSTRACT

Transferability estimation metrics are used to find a high-performing pre-trained model for a given target task without fine-tuning the models and without access to the source dataset. Despite the growing interest in developing such metrics, the benchmarks used to measure their progress have largely gone unexamined. In this work, we empirically show the shortcomings of widely used benchmark setups for evaluating transferability estimation metrics. We argue that the benchmarks on which these metrics are evaluated are fundamentally flawed. We empirically demonstrate that their unrealistic model spaces and static performance hierarchies artificially inflate the perceived performance of existing metrics to the point where simple, dataset-agnostic heuristics can outperform sophisticated methods. Our analysis reveals a critical disconnect between current evaluation protocols and the complexities of real-world model selection. To address this, we provide concrete recommendations for constructing more robust and realistic benchmarks to guide future research in a more meaningful direction as well as provide a benchmark based on the proposed best practices for vision classification.

## 1 INTRODUCTION

Using models that are pre-trained on large datasets like ImageNet (Deng et al., 2009) has become standard practice in real-world deep-learning scenarios. However, performance gains can vary considerably depending on the model architecture, weights, and the dataset on which it was pre-trained (the source dataset). This leads to the pre-trained model selection problem. This raises the question: *"How can we find a high-performing pre-trained model for a given target task without fine-tuning our models and without access to the source dataset."* Source Independent Transferability Estimation (SITE) metrics address this question by computing a cheap-to-calculate score for each candidate model, that is used to rank models by their predicted downstream performance. This research area is growing rapidly, with papers appearing at major AI venues such as ICML, NeurIPS, and CVPR. Progress in the field has so far been measured primarily against a small set of widely adopted benchmarks.

While these benchmarks have been useful for driving early advances, we argue that they fail to capture the complexities of real-world applications. To address this gap, our paper offers a critical empirical analysis of the most commonly used benchmark setup for SITE metric evaluation. We identify fundamental flaws in its design that call into question the validity of reported results. Our contributions are to:

1. Empirically demonstrate how current benchmarking practices give misleading performances of SITE metrics which questions their reliability,

2. Show that a simple, static ranking heuristic can outperform sophisticated metrics, exposing the trivial nature of the task posed by the benchmark which further highlights the weakness of the current benchmark, and

3. Propose a set of actionable best practices for constructing more robust and meaningful benchmarks suitable for practical challenges of real-world model selection.

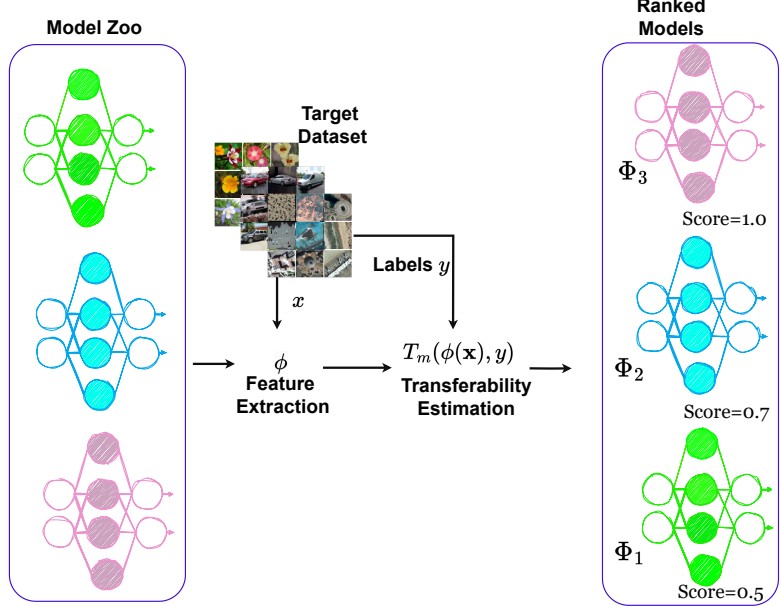

Figure 1: Illustration of Source Independent Transferability Estimation (SITE): Given a set of pre-trained models (on the left), a SITE metric computes a score $T_m$ based on extracted features on a target dataset. The scores $T_m$ are used to rank the pre-trained models according to their transferability.

Despite significant progress in the development of novel SITE methods, the empirical evaluation standards within this emerging research area have yet to reach the maturity seen in other machine learning domains. To address this issue, we propose best practices to rectify them, leading to the SITE benchmarking and evaluation checklist (Appendix A).

## 2 TRANSFERABILITY ESTIMATION: BACKGROUND

The idea behind transferability estimation is simple: estimate which model from a zoo of models would perform best after fine-tuning. Transferability estimation as a research area is fairly young; the H-Score (Bao et al., 2019) and NCE (Tran et al., 2019) can be considered early works on this topic, introducing the evaluation of transferability and the assignment of models corresponding to an estimate of their transferability for a given target task. We illustrate how transferability estimation metrics work in Figure 1.

There are two widely accepted problem scenarios for transferability estimation: source-dependent transferability estimation (where one has access to the source and target dataset) and source-independent transferability estimation (where one does not have access to the source dataset), we will be focusing on the latter.

### 2.1 SOURCE DEPENDENT TRANSFERABILITY ESTIMATION (SDTE)

The SDTE scenario assumes access to the source datasets on which the models have been pre-trained. Apart from the fact that this assumption is often not met, a drawback of common SDTE metrics is the use of distribution matching methods like optimal transport (Tan et al., 2021), which are typically very expensive to compute. In addition, SDTE metrics are not reliable when the discrepancy between the source and target datasets is very high(Braccaioli et al., 2025); for example, when comparing the entire ImageNet21K (Deng et al., 2009) to the Cars (Krause et al., 2013) or Plants (G. & J., 2019) datasets.

## 2.2 Source Independent Transferability Estimation (SITE)

The Source Independent Transferability Estimation (SITE) assumes access to the source model but not the source training data. This is a more realistic scenario, as we might not always have access to the source dataset, nor have the capacity to store the typically very large source datasets like ImageNet (Deng et al., 2009) or LAION (Schuhmann et al., 2022) in our local setup. SITE methods typically rely on evaluating the feature representation of the source model on the target dataset and its relationship with the target labels. There are several SITE metrics inspired by various viewpoints. For instance, LogME (You et al., 2021) formalizes the transferability estimation as the maximum label marginalized likelihood and adopts a directed graphical model to solve it. SFDA (Shao et al., 2022) proposes a self-challenging mechanism; it first maps the features and then calculates the sum of log-likelihood as the metric. ETran (Gholami et al., 2023) and PED (Li et al., 2023) treat the problem of SITE with an energy function and act as a pre-processor for other SITE methods. ETran uses energy-based models to detect whether a target dataset is in-distribution or out of distribution for a given pre-trained model, whereas PED utilizes a potential energy function to modify feature representations to aid other transferability metrics like LogME and SFDA. LEEP (Nguyen et al., 2020) is the average log-likelihood of the log-expected empirical predictor, which is a non-parametric classifier based on the joint distribution of the source and target distribution. N-LEEP (Li et al., 2021) is a further improvement on LEEP by substituting the output layer with a Gaussian mixture model. TransRate (Huang et al., 2022) treats SITE from an information theory point of view by measuring the transferability as the mutual information between features of target examples extracted by a pre-trained model and their labels. There have been applications of transferability estimation and proposed metrics for different domains such as medical imaging (Juodelyte et al., 2024; Singh et al., 2025; Yang et al., 2023) and speech recognition (Chen et al., 2023). We suggest the survey by Ding et al. (2024) for a complete view of transferability metrics.

## 3 SITE: Problem Statement

We assume that we are given a target dataset $\mathcal{D} = \{(\mathbf{x}_n, y_n)\}_{n=1}^{N}$ of $N$ labeled points and $M$ pre-trained models $\{\Phi_m = (\phi_m, \psi_m)\}_{m=1}^{M}$. Each model $\Phi_m$ consists of a feature extractor $\phi_m(x) \in \mathbb{R}^d$ that returns a $d$-dimensional embedding and the final layer or head $\psi_m$ that outputs the label prediction for the given input $\phi_m(x)$. The task of estimating transferability is to generate a score for each pre-trained model so that the best model is at the top of a ranking list. For each pre-trained model $\Phi_m$, a transferability metric outputs a scalar score $T_m$ that should be coherent in its ranking with the performance of the fine-tuned classifier $\hat{\Phi}_m$. That is, the goal is to obtain scores such that score $T_{m_1} \geq T_{m_2}$ if and only if the fine-tuned model $\hat{\Phi}_{m_1}$ has a higher probability to predict the correct labels on the target dataset than model $\hat{\Phi}_{m_2}$:

$$\frac{1}{N} \sum_{n=1}^{N} p(y_n | \mathbf{x}_n; \hat{\Phi}_{m_1}) \geq \frac{1}{N} \sum_{n=1}^{N} p(y_n | \mathbf{x}_n; \hat{\Phi}_{m_2}),$$

where $p(y_n | x_n; \hat{\Phi}_m)$ indicates the probability that the fine-tuned model $\hat{\Phi}_m$ predicts label $y_n$ for input $\mathbf{x}_n$. Hence, a larger $T_m$ should correspond to a better performance of the model on target data $\mathcal{D}$.

## 4 Transferability Estimation: Standard setup

Widely adopted transferability estimation methods, such as LogME (You et al., 2021), TransRate (Huang et al., 2022), NCTI (Wang et al., 2023), LEEP (Nguyen et al., 2020), SFDA (Shao et al., 2022), ETran (Gholami et al., 2023), GBC (Pándy et al., 2022) and LEAD (Hu et al., 2024) are evaluated on benchmarks sharing similar models and datasets.

The setup used in LogME, TransRate, H-Score, SFDA, ETran, and NCTI uses pre-trained ResNets (He et al., 2015) (ResNet34, ResNet50, ResNet101, ResNet151), DenseNets (Huang et al., 2016) (DenseNet169, DenseNet121, DenseNet201), MobileNet (Howard et al., 2017), Inceptionv3 (Szegedy et al., 2015), MNASNet (Tan et al., 2018) and GoogleNet (Szegedy et al., 2014). These models are fine-tuned on CIFAR10 (Krizhevsky et al., 2009), CIFAR100 (Krizhevsky et al., 2009), Pets (Parkhi

et al., 2012), Aircraft (Maji et al., 2013a), Food (Bossard et al., 2014) and DTD (Cimpoi et al., 2014). In this work, we focus on this widely adopted SITE benchmark setup.

**Fine-tuning details**

To obtain test accuracies, these models were fine-tuned with a grid search over learning rates $\{10^{-1}, 10^{-2}, 10^{-3}, 10^{-4}\}$ and a weight decay in $\{10^{-3}, 10^{-4}, 10^{-5}, 10^{-6}, 0\}$. The best hyperparameters are determined based on the validation set. The final model is fine-tuned on the target dataset with the selected parameters. The resulting test accuracy is used as the ground truth score $G_m$ for model $\Phi_m$. This way, we obtain a set of scores $\{G_m\}_{m=1}^M$ as the ground truth to evaluate our pre-trained model rankings.

**Evaluation Protocol**

The evaluation of transferability metrics reflects the correlation between the ground truth accuracy and the achieved SITE score. Currently, weighted Kendall's Tau (Vigna, 2015) prevails as a measure to estimate the correlation. Earlier transferability works like H-score used Pearson correlation as an evaluation metric, but Pearson's $r$ is considered as too sensitive to scale, as two scorings will induce the same order can differ in their evaluation merely due to calibration. Kendall's $\tau$ is a more interpretable rank statistic, as it counts the number of swaps that bubble sort would have to perform to put one list in the same order as the other one. More precisely, Kendall's $\tau$ returns the ratio of concordant pairs minus discordant pairs when enumerating all pairs of transferability estimations $\{T_m\}_{m=1}^M$ and ground truth transferability scores $\{G_m\}_{m=1}^M$, as given by:

$$\tau = \frac{2}{M(M-1)} \sum_{1 \leq i < j \leq M} \text{sgn}(G_i - G_j)\,\text{sgn}(T_i - T_j).$$

The sgn function returns the sign of the input value and zero if the input value is zero. Since practical applications rely mainly on the correct order for the top performing pre-trained models, a weighted variant of Kendall's $\tau_w$ is used. Weighted Kendall's $\tau_w$ is computed as

$$\tau_w = \frac{2}{M(M-1)} \sum_{1 \leq i < j \leq M} \text{sgn}(G_i - G_j)\,\text{sgn}(T_i - T_j)w(\rho(i), \rho(j)),$$

where $\rho(i)$ returns a ranking of indices, starting at zero. As a default choice, weighted Kendall's tau uses hyperbolic weighing : $w(r, s) = \frac{1}{s+1} + \frac{1}{r+1}$. In transferability estimation, the ranking $\rho$ reflects the ranking of the ground truth accuracy. As a result, any pair that involves one of the top transferring models gets a large weight. In the rest of the text, we will refer to this benchmark (datasets, models, evaluation protocol) as *standard benchmark*.

## 5 AN EMPIRICAL CRITIQUE OF THE STANDARD BENCHMARK

While the standard benchmark is widely used, we contend that it is built on flawed foundations that lead to an overestimation of the true capabilities of SITE metrics. We identify and empirically validate three critical limitations: an unrealistic model space that does not reflect practical challenges in transferability estimation, a benchmark that is solved by a static ranking for all considered datasets, and misleading differences in SITE scores that do not meaningfully correlate with performance gaps. For the purpose of this study, we examine the following metrics on the standard benchmark: LogME (You et al., 2021), TransRate (Huang et al., 2022), GBC (Pándy et al., 2022), NLEEP (Li et al., 2021), (Shao et al., 2022), and H-Score (Bao et al., 2019)

### 5.1 CRITIQUE 1: THE MODEL SEARCH SPACE IS UNREALISTIC

We argue that the model space of the standard benchmark's model pool is **unrealistic**, because it is dominated by models of varying sizes from only two architectural families (ResNets and DenseNets). In a real-world scenario, practitioners are not interested in knowing whether they should use a "bigger vs. smaller" ResNet, but rather which architecture performs under particular constraints such as size , training speed, inference speed, and ease of access. For example, larger vision classification models are known to predictably outperform their shallower counterparts (He et al., 2015), and mixing small and large variants of the same family of vision classification models reduces the complex task of model selection to a trivial detection of the largest model.

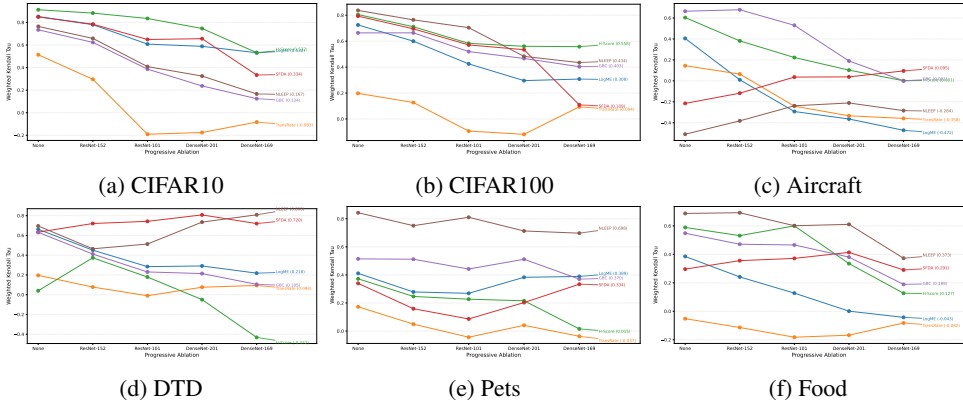

Figure 2: Performance of transferability metrics when we remove architectures of the same families. We sequentially remove the architectures denoted on the horizontal axis and report the achieved $\tau_w$. A pattern of performance decrease with every ablation is observed.

The inclusion of MobileNet and MNASNet alongside ResNets and DenseNets is likewise misaligned with the benchmark's use case. MobileNet and MNASNet are designed for edge computing environments. In the experimental evaluation, MobileNet and MNASNet consistently occupy the bottom ranks (cf. Figure 3). Hence, removing those models from the search space has little effect on the performance evaluation. However, we advocate against using these models in the standard benchmark to avoid unnecessary inflation of the search space.

We therefore recommend (i) excluding edge-oriented models from the benchmark and (ii) using at most one representative architecture per family, ensuring that the compared models have similar sizes.

**Validation via Model Ablation** To test the robustness of SITE metrics under a more realistic search space, we ablate the largest models from overrepresented families (ResNet-152, ResNet-101, DenseNet-169, DenseNet-201) and recompute weighted Kendall's $\tau_w$. Figure 2 plots $\tau_w$ as we progressively remove these models, indicated on the horizontal axis. The rightmost points correspond to a setting with one model per family, containing only 7 of the initial 11 models.

Most SITE metrics exhibit a sharp drop in $\tau_w$ after ablation. Except on DTD and Pets, all SITE metrics fall below $0.6$ once the oversized variants are removed. The results also show that none of the metrics are robust to changes in the model space. For every metric, a dataset exists where the removal of a single model results in a steep decrease in performance. In this more realistic setup, no metric reliably predicts transferability across datasets. This demonstrates that the high performance of these metrics is brittle and heavily reliant on correctly ranking a few over-represented models in a flawed benchmark. Most importantly, none of the existing SITE metrics are able to reliably estimate transferability in a setup where models of similar sizes are compared.

## 5.2 CRITIQUE 2: THE BENCHMARK IS SOLVED BY A STATIC RANKING

Dependencies between the candidate models (having larger and smaller variants of the same architecture in the model search space) and a lack of diversity in the evaluated datasets lead to a **static leaderboard**, where a few high-capacity models, such as ResNet-152, consistently occupy the top ranks regardless of the target dataset. We visualize the ranking of models in Figure 3. The figure shows that the top performing model, ResNet-152, occupies the first rank for 8 of the 10 datasets in the benchmark. The second place is always occupied by one of the top 3 models (ResNet-152, DenseNet-201, and ResNet-101).

As a result, we question whether the standard benchmark is suitable for assessing transferability estimation if a SITE metric with a data-independent ranking can achieve high-performance measurement. We probe this question by introducing a controlled experiment that tests whether a fixed, dataset-agnostic ranker can rival SITE metrics. This leads us to validate the hypothesis through

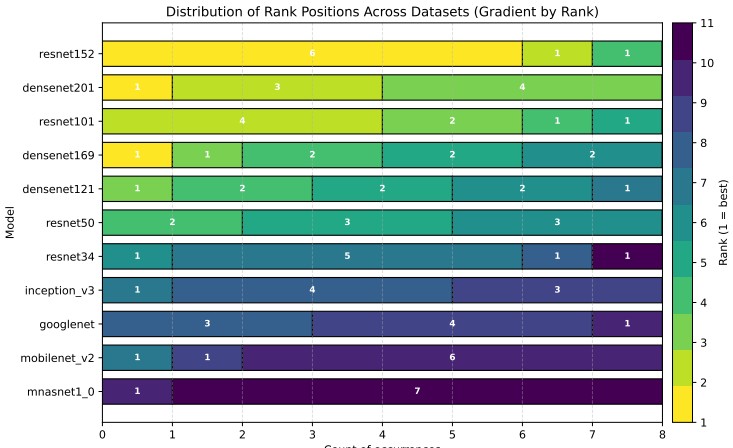

Figure 3: Visualization of the ranking distribution of models in the standard benchmark(ordered by fine tuned performance). The models at the top occupy the first ranks in most datasets.

| Dataset Metric | Aircraft | CIFAR10 | CIFAR100 | DTD | Food | Pets | Average |
|---|---|---|---|---|---|---|---|
| GBC | -0.12 | -0.02 | 0.09 | 0.14 | 0.10 | -0.15 | 0.007 |
| TransRate | 0.14 | 0.51 | 0.20 | 0.20 | -0.05 | 0.17 | 0.195 |
| SFDA | -0.22 | 0.85 | 0.79 | 0.63 | 0.30 | 0.34 | 0.448 |
| H-Score | 0.60 | 0.91 | 0.80 | 0.04 | 0.59 | 0.37 | 0.552 |
| NLEEP | -0.51 | 0.76 | 0.84 | 0.70 | 0.69 | 0.84 | 0.553 |
| LogME | 0.41 | 0.85 | 0.72 | 0.66 | 0.39 | 0.41 | 0.573 |
| Static Ranking | **0.84** | **0.91** | **0.98** | **0.99** | **0.80** | **0.94** | **0.91** |

Table 1: Comparison of transferability estimations, computed by weighted Kendall's tau, for a static ranking versus SITE metrics on the standard benchmark. The static ranking achieves the highest $\tau_w$ overall.

a static ranker that exploits the leaderboard's inherent bias without leveraging any task-specific information.

**Validation via a Static Ranking Heuristic**   We test a naive **static ranker** in the standard benchmark, which orders models according to a fixed sequence. The static ranking follows a simple heuristic based on the model size and an alternation of ResNet and DenseNet model families, resulting in the following order:

ResNet-152≻DenseNet-201≻ResNet-101≻DenseNet-169≻ResNet-50≻DenseNet-121≻ResNet-34≻GoogleNet≻Inception3≻MobileNet≻MNASNet.

We report the performance of this static ranking in Table 1. We observe that the static ranking significantly outperforms the sophisticated SITE metrics, achieving the highest $\tau_w$ on every dataset. On average, the static ranking achieves $\tau_w = 0.91$, while the best performing SITE metric, LogME, achieves $\tau_w = 0.57$. This finding questions the insight gained from the standard benchmark, as it rewards the memorization of a fixed model hierarchy rather than the ability to perform true task-specific transferability estimation.

## 5.3   CRITIQUE 3: SITE METRICS ARE NOT EVALUATED TOWARDS FIDELITY

Beyond ranking, a practical transferability metric should provide scores whose *magnitudes* are meaningful. That is, a large gap in metric scores should correspond to a large gap in downstream accuracy, allowing a user to assess whether selecting a higher-scoring model is worth the potential

increase in computational cost. The standard benchmark evaluation protocol, focused solely on rank correlation, overlooks this crucial property.

We formalize this property as follows. Let $\Delta_{\text{Acc}}$ be the difference in accuracy between two models on a target dataset $\mathcal{D}$, and let $\Delta_T$ be the difference in their transferability scores:

$$\Delta_{\text{Acc}}(X, Y; \mathcal{D}) = \text{Acc}(X, \mathcal{D}) - \text{Acc}(Y, \mathcal{D}), \quad \Delta_T(X, Y) = T(X) - T(Y)$$

An ideal metric should preserve the ordering of differences: for any four models $A, B, C, D$ from the model space $\mathcal{M}$:

$$\forall A, B, C, D \in \mathcal{M}, \quad \Delta_{\text{Acc}}(A, B; \mathcal{D}) > \Delta_{\text{Acc}}(C, D; \mathcal{D}) \implies \Delta_T(A, B) > \Delta_T(C, D)$$

**Validation via Pairwise Difference Correlation**   To quantify this property, which we term *fidelity to accuracy differences*, we formalize it as the correlation between pairwise differences in accuracy ($\Delta_{\text{Acc}}$) and pairwise differences in the metric's score ($\Delta_T$). A high correlation would indicate that score differences are meaningful proxies for performance gaps.

We compute the Pearson correlation between all $\{\Delta_{\text{Acc}}\}$ and $\{\Delta_T\}$ pairs for each metric and dataset. The resulting heatmap is shown in Figure 4. We observe that nearly all metrics exhibit a weak correlation with accuracy differences. For example, our analysis reveals that a LogME score difference of 0.09 can correspond to an accuracy gap as large as 2.5% or as small as 0.5% in the Pets dataset. A more detailed plot of the relationships from scores to ground truth accuracies can be found in Appendix E. The lack of a reliable mapping between score gaps and performance gains severely limits the practical utility of these metrics for end-users, who cannot confidently interpret the scores to make informed decisions.

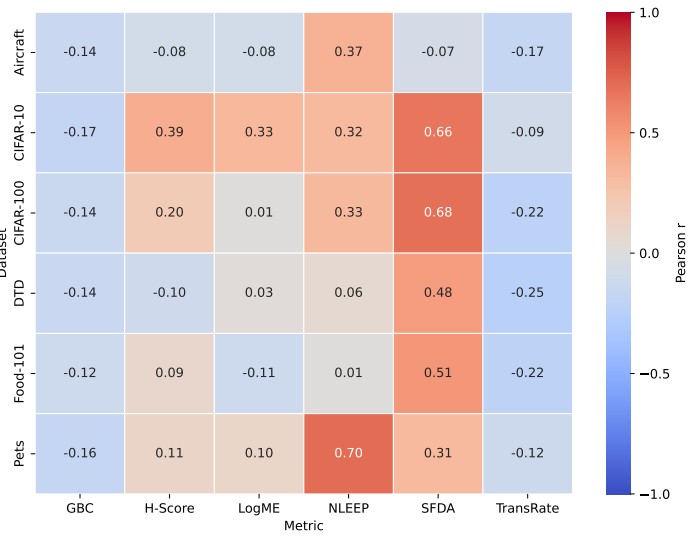

Figure 4: Heatmap correlation of $\Delta_{Acc}$ and $\Delta_T$

## 5.4 Transferability Estimation Outside of Computer Vision

In this work, we focus on popular computer vision and image classification setups; however, but other fields have similar benchmark issues. For example, MEAF (Hao & Wang, 2025) is a SITE metric for spiking neural networks, where our critique applies. The SEW-ResNet-152 model dominates the top rankings, and the performance differences can be as small as 0.2% at the second rank (which can occur due to fine-tuning differences). LogME experiments on NLP tasks also reveal one static winner every time as shown in Table 4. Critique 2, the comparison of models with varying sizes from the same families, can also be applied to recent work in object detection Zhang et al. (2025). Here, varying sizes of YOLO models were compared to one another for the same task, in which YOLOv5m won in 4 out of 5 tasks. EMMS (Meng et al., 2023) conducts an investigation into the transferability of ViT models while comparing ViT-S to ViT-B, where ViT-B outperforms every model and holds the first rank in 8 out of 11 tasks.

## 6 RELATED WORK

Previous work has conducted large-scale analysis of Source Independent Transferability Estimation (SITE) metrics. For instance, Agostinelli et al. (2022) perform a comprehensive evaluation across over 700,000 experiments, showing that the effectiveness of a transferability metric is highly dependent on the specific experimental scenario. Although the setup proposed by Agostinelli et al. (2022) is thorough, its scale makes it impractical to reproduce in typical research settings without substantial computational resources. Similarly, Ibrahim et al. (2021) highlight the instability of SITE metrics in the presence of class imbalance and propose an adaptation of the H-score tailored to their setting. Baker & Handmann (2024) recommend different SITE metrics for different scenarios.

Chaves et al. (2024), Kazemi et al. (2025), and Bolya et al. (2022) mention some drawbacks of the standard benchmark but continue to utilize an impractical model search space (Critique 1) and may encounter similar challenges as those associated with a static leaderboard (Critique 2). Kornblith et al. (2019) show that higher ImageNet accuracy predicts better transfer on web-scraped target datasets, which largely overlap with the datasets of the standard benchmark. Fang et al. (2023)) demonstrate that this correlation breaks down for real-world datasets that are not web-scraped. Our work complements these findings by exposing dataset and model search space synergies that result in benchmark limitations.

In contrast to these studies, our goal is not to evaluate a particular SITE metric or to recommend which method to use and when, but rather to examine the benchmarks and recommend best practices for evaluating SITE metrics. We introduce a framework for assessing the effectiveness of SITE benchmarks and offer practical insights into constructing experimental setups that enable more reliable evaluations and better inform the practitioner, as the end use of these metrics is to provide improved evaluations for users.

## 7 DISCUSSION

In this section, we move from critique to construction and propose a set of actionable recommendations designed to foster the development of more robust and practically relevant benchmarks. We show how we can address the limitations identified in the previous section and move towards building a more reliable benchmark setup for SITE metrics. The goal is to ensure that future transferability metrics are validated against challenges that mirror the complexities faced by practitioners, thereby providing trustworthy guidance for real-world model selection. We therefore polish recommendations now on "How to build better benchmarks for your transferability estimation metric."

**Best Practice 1: Release code and data for your transferability estimation metric.**

In this work, we were only able to examine a handful of metrics and datasets due to the limitations of publicly available data. A good practice to ensure reproducibility and reduce noise in score discrepancies is to release the code for the metric, the datasets with links on which it was trained, the score metrics obtained, final accuracies, and the pretrained models on which the score was computed (for different task transfer scenarios). This, firstly, ensures full reproducibility and, secondly, allows for the incorporation of inconsistent scores in different works for the same task.

**Best Practice 2: Construct a Diverse and Non-Trivial Model Space**

To create a more robust evaluation, the model space must be intentionally diversified to present a non-trivial selection problem. This involves curating a set of models from **architecturally distinct paradigms**, such as Convolution Neural Networks (e.g., ConvNeXt), Vision Transformers (e.g., ViT, Swin), and MLP-based models (e.g., MLP-Mixer). Furthermore, to ensure a fair comparison that isolates the impact of architectural inductive bias, comparable models should be chosen according to practitioners constraints, such as computational budgets (e.g., parameter counts or FLOPs), inference speed, training speed, etc. This forces the evaluation of the metric to move beyond simple scaling rules and to make nuanced judgments about which architecture is best suited for the specific downstream task.

**Best Practice 3: Ensuring a Diverse and Challenging Dataset Space**

The suite of downstream tasks is as critical as the model space. A metrics utility is defined by its ability to generalize across a wide range of applications. A benchmark that relies on a narrow set of

similar or overly simplistic datasets cannot provide this assurance. We identify two essential axes of diversity for the dataset space:

- **Task Difficulty and Performance Headroom:** Many existing benchmarks use datasets where modern pre-trained models achieve near-saturated performance (e.g., 99% accuracy). This "performance ceiling" makes meaningful rankings impossible, as the marginal differences between top models are often statistically insignificant and fall within the noise of the fine-tuning process. A robust benchmark must include challenging datasets that provide sufficient performance headroom for even the strongest models, ensuring that a clear and statistically significant performance gap can be measured.

- **Domain and Task Variety:** A metric's robustness is tested by its performance across varied domains and task types. Consequently, the benchmark ought to incorporate datasets that cover a variety of visual domains, including fine-grained classification (for instance, FGVC Aircraft (Maji et al., 2013b), Stanford Cars (Krause et al., 2013)), medical imaging, satellite imagery, and texture analysis (such as DTD Cimpoi et al. (2014)). Additionally, datasets that are not sourced from the internet should be included, consistent with the findings of Fang et al. (2023). This evaluates the metric's capacity to handle varying degrees of domain shift from the original source pre-training data. We acknowledge that the property of diverse datasets is generally present in the majority of the SITE papers.

**Best Practice 4: Engineering for Performance Spread and Rank Dispersion**

A critical but often overlooked flaw in standard benchmarks is the persistence of a static model hierarchy, where one or two models dominate nearly all tasks. This "static leaderboard" undermines evaluation validity: a transferability metric can achieve a high weighted Kendall's Tau simply by favoring the top model rather than by accurately predicting task-specific suitability. To isolate architectural inductive bias, models should be matched on computational budgets (e.g., parameter counts or FLOPs). This design forces transferability metrics to move beyond simple scaling laws and, instead, to make nuanced judgments about which architecture best fits a given task.

Equally important, an effective benchmark must exhibit high rank dispersion: model rankings should vary substantially across tasks, with different architectures excelling on different datasets. Achieving this requires a careful co-design of the model pool and dataset pool so that unique inductive biases are rewarded in different contexts. Such a setup provides a far more rigorous test of transferability metrics, ensuring they identify the right model for the right task rather than defaulting to the same top performer.

We view these guidelines as a first step toward evolving standards for benchmarking transferability estimation, which may need to adapt to different deployment settings, such as edge devices versus large-scale computing environments. While our guidelines are essential steps, we also provide a benchmark for image classification that takes into account Best Practices 2,3, and 4 in the next section.

## 8 BENCHMARK BASED ON BEST PRACTICES

We built a robust benchmark based on the best practices described in the previous section. Our proposed benchmark consists of the following models: Twins-SVT (Chu et al., 2021), XCiT (El-Nouby et al., 2021), CoaT (Xu et al., 2021), DeiT (Touvron et al., 2021), MaxViT (Tu et al., 2022), and MViT v2 (Li et al., 2022). This selection covers complementary model design paradigms, while all models have a similar parameter range, and no model is a direct improvement over the other. Fine-tuning these selected models on datasets from the *standard benchmark* results in multiple models obtaining $> 99\%$ accuracy, which is not meaningful for SITE. To address this limitation, we expand the benchmark's datasets with selected datasets from the Meta-Album (Ullah et al., 2022). From the 30 datasets of the Meta-Album, we do not select those where multiple models achieve very high or close to 100% accuracy.

Our resulting benchmark consists of the following datasets: Sports (Piosenka), Plant Village (G. & J., 2019), RESISC (Cheng et al., 2017), Insects (Wu et al., 2019), PanNuke (Gamper et al., 2019), MPII Human Pose Dataset (Andriluka et al., 2014),Fungi (Picek et al., 2021), RSD (Long et al., 2017), Boats (Gundogdu et al., 2017), Plant Doc (Singh et al., 2020), Stanford Actions (Yao et al., 2011),

DTD (Cimpoi et al., 2014), Subcellular Human Protein Dataset(PRTA) (Thul et al., 2017), Insects SPIPOLL (Wu et al., 2019), and Dogs Khosla et al. (2011). Based on Critique 2, we evaluate our benchmark against a static ranker. We create this static ranker based on the fine-tuned performance of models; more information on this can be found in the Appendix B.

We show the performance of SITE metrics and the static ranker on our benchmark in Table 2. We observe that not a single SITE metric can perform consistently well for our benchmark. The static ranker achieves comparably low values of $\tau_w \in [-0.3, 0.77]$ with a mean of 0.31. While following best practices, we propose that this benchmark should be used as an example to build a better and more reliable benchmark for SITE metric evaluation for different tasks, such as computer vision, NLP, and information retrieval.

| Dataset | TransRate | LogME | NLEEP | SFDA | HScore | GBC | **Static** |
|---|---|---|---|---|---|---|---|
| Sports | 0.39 | 0.25 | 0.30 | **0.70** | -0.08 | 0.38 | 0.46 |
| PlantVillage | 0.18 | **0.61** | **0.61** | -0.05 | 0.30 | 0.14 | -0.30 |
| RESISC | 0.24 | 0.11 | 0.14 | **0.76** | 0.23 | 0.36 | 0.55 |
| Stanford Actions | -0.16 | -0.37 | -0.28 | -0.07 | 0.01 | 0.03 | **0.27** |
| Insects | 0.72 | 0.57 | 0.84 | 0.53 | 0.52 | **0.87** | 0.56 |
| DTD | -0.53 | -0.37 | -0.37 | -0.48 | -0.33 | -0.42 | **0.01** |
| PanNuke | 0.14 | -0.06 | 0.24 | **0.68** | 0.13 | 0.40 | 0.00 |
| Dogs | -0.71 | -0.41 | -0.62 | -0.32 | -0.30 | -0.59 | **-0.15** |
| MPII Human | 0.34 | 0.27 | 0.25 | 0.18 | 0.24 | 0.23 | **0.50** |
| Fungi | 0.44 | 0.70 | -0.22 | **0.77** | 0.40 | 0.34 | **0.77** |
| Plant Doc | 0.54 | 0.30 | 0.30 | 0.10 | 0.10 | 0.48 | **0.58** |
| SPIPOLL | -0.15 | 0.10 | -0.15 | -0.28 | -0.18 | -0.15 | **0.32** |
| RSD | 0.02 | -0.20 | -0.34 | -0.07 | -0.06 | -0.04 | **0.60** |
| PRTA | 0.28 | -0.09 | -0.44 | 0.14 | **0.38** | 0.29 | 0.37 |
| Boats | 0.20 | -0.49 | **0.38** | -0.32 | -0.33 | 0.27 | 0.09 |
| Average | 0.13 | 0.061 | 0.04 | 0.15 | 0.06 | 0.17 | **0.31** |

Table 2: $\tau_w$ performance on 15 benchmark datasets from Meta-Album, showing the performance of LogME, SFDA,HScore,GBC, NLEEP, TransRate and Static Ranker.

## 9 LIMITATIONS AND FUTURE WORK

Our work examines the standard SITE benchmark for image classification; our investigation can serve as a blueprint for future studies in other setups and can be further expanded to NLP, object detection, and medical image classification domains as well. A limitation that current benchmarks and metrics suffer from is the integration of different finetuning strategies, optimizers, and hyperparameters in the SITE metric evaluation. Currently, the methods and benchmarks have not been developed to take these hyperparameters into account while predicting transferability, whereas research has shown that they play a significant role in model performance. A promising direction can be the incorporation of learning from social choice theory (Zhang & Hardt, 2024) to improve the development of more reliable benchmarks for SITE.

## 10 CONCLUSION

Benchmarking is the cornerstone of machine learning research, allowing researchers to develop robust ideas and enabling scientific progress. Without robust benchmarks, trust in research methods can erode; to prevent this, we have shown how to ground benchmarks in more realistic scenarios for SITE metrics. Our experiments highlighted critical failures in current benchmarking practices. To aid future research, we provide actionable best practices and a SITE benchmark and evaluation checklist for constructing robust benchmarks in the Appendix A, inspired by the NAS Checklist (Lindauer & Hutter, 2020). Our set of recommendations and proposed experiments is actionable and concrete, ensuring robust benchmarking of SITE metrics. This work serves as a call to action for the community to adopt more rigorous standards; by doing so, we can foster the development of transferability metrics that are genuinely useful to practitioners and provide truly predictive and reliable guidance.

## 11 ACKNOWLEDGEMENTS

This work was supported by European Union's Horizon Europe research and innovation programme under grant agreement number 101214398 (ELLIOT).

## 12 REPRODUCIBILITY STATEMENT

To enable reproducibility, we provide the code, data, and execution scripts in the supplementary files. We also provide additional code for our experiments and Jupyter notebooks to reproduce the figures.

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

# Supplementary Material

## A    SITE Benchmark and Evaluation Checklist

To aid future research, we provide a concrete checklist for constructing robust benchmarks inspired by NAS Checklist (Lindauer & Hutter, 2020)

**Best practices for Building benchmark.**

- ❏ Ensure Diversity (Models should be from different families).
- ❏ Match computational budgets (Models should be under similar parameter range).
- ❏ Avoid Trivial hierarchies (Models should not be in incremental improvements over each other eg Inceptionv2, Inceptionv3, Inceptionv4) where one improvement is a proven improvement over other.
- ❏ Include datasets with room for improvement. (Do not include datasets where all scores ¿ 99%).
- ❏ Include datasets from multiple domains.
- ❏ Engineer for rank dispersion (so different models win in different tasks) to avoid a static leaderboard. If this is not possible then examine if the following task requires transferability estimation.

**Best practices for reporting experiments and evaluation**

- ❏ Report $\tau_w$ with ablation over every model.
- ❏ Report correlation of $\Delta_T$ and $\Delta_{Acc}$

**Best practices for releasing code** For all experiments you report, check if you released:

- ❏ Code for the training pipeline used to evaluate the final architectures.
- ❏ Code for computing SITE metric for specialized tasks as well like object detection and Regression.

## B    BENCHMARK BASED ON BEST PRACTICES

### B.1    STATIC RANKER

The final static ranking is determined by sorting the models based on these histograms. A model is ranked higher if it has more first-place finishes. If two models have the same number of 1st place finishes, the tie is broken by comparing their number of 2nd place finishes. This process continues lexicographically through all rank positions, ensuring a stable and deterministic ordering.

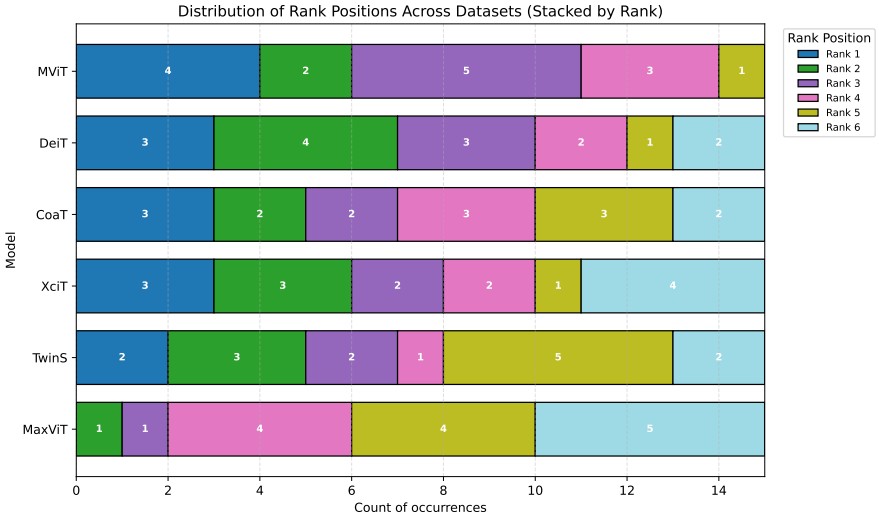

Figure 5: Visualization of the ranking distribution of models in the improved benchmark. Most of the models share top spots and no model is always top or bottom rank.

## C    ADDITIONAL ANALYSIS

### C.1    TOP-K MODELS ANALYSIS

The practical importance of fidelity is most pronounced when comparing top-performing models. We therefore conducted a focused analysis on model pairs involving at least one from the top-3 ranks. The results show that even in this critical subset, the correlation between score gaps ($\Delta_T$) and accuracy gaps ($\Delta_{Acc}$) remains weak and unreliable across all metrics. Large differences in SITE scores between top-ranked models frequently corresponded to insignificant gaps in fine-tuned accuracy, confirming that these metrics fail to provide meaningful guidance for selecting the best model. This lack of fidelity for the most relevant models underscores their limited practical value for real-world decision-making. We provide a detailed breakdown of this analysis in Figure 6 and Figure 7.

### C.2    AGGREGATED PEARSON CORRELATION FOR METRIC FIDELITY

In Table 3, we list the average Pearson correlation of every metric between $\Delta_T$ and $\Delta_{Acc}$, this metric can be useful for practitioners to get a average idea of fidelity of different SITE metrics.

## D    LISTING FLAWED TABLES FROM PREVIOUS STUDIES

In Table 4 we can also observe that the RoBERTa model always outperforms other models, and BERT-D always underperforms, which causes similar biases as discussed in our work. We observe similar benchmark practices here, such as standard benchmarks with multiple different parameter ranges of models coupled together and a static leaderboard.

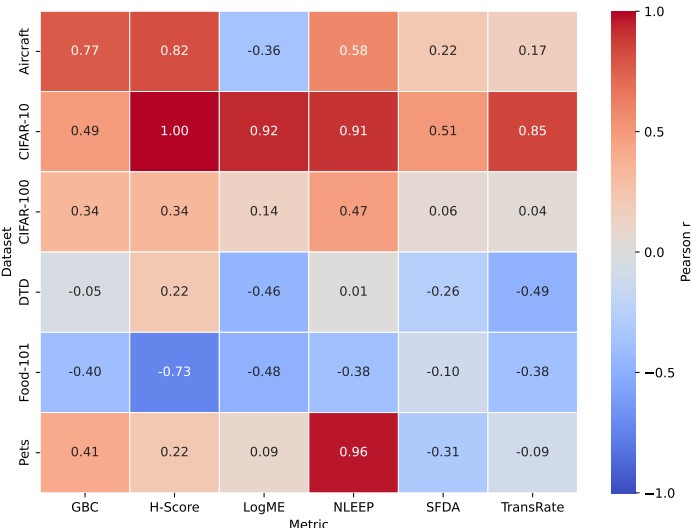

Figure 6: Heatmap correlation of $\Delta_{Acc}$ and $\Delta_T$ for Top-4 Models

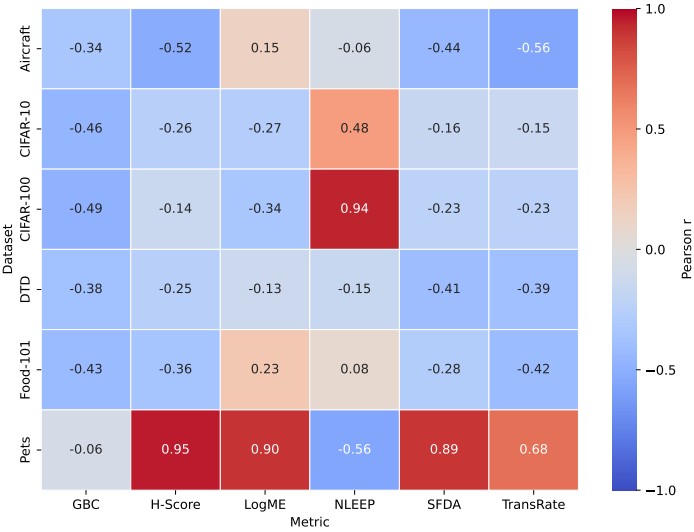

Figure 7: Heatmap correlation of $\Delta_{Acc}$ and $\Delta_T$ for Bottom-4 models.

# E    SCORE VS ACCURACY FOR ALL SITE METRICS

We plot the SITE scores against the achieved ground truth accuracy to provide a more detailed picture of the fidelity of the scores to the differences in accuracy. Ideally, we observe a linear relationship such that we can infer from a large gap between scores a (comparatively) large gap between accuracies.

Figure 8 plots the GBC score against the accuracy. We observe that the distances between scores have no meaningful translation to the obtained accuracies. The majority of scores cluster in a very small range, and some outliers obtain visibly distinctive scores. Figure 9 shows that LogME is able to reflect meaningful distances in accuracy with its score for the CIFAR-10, and Cifar-100 and Food-101 datasets at least for the top performing models. This tendency is also observed in the correlation heatmap (cf. Figure 4). NLEEP indicates somewhat linear relationships for the CIFAR-10, CIFAR-100, Pets, and Food-101 dataset (cf. Figure 10). For the Aircraft dataset we observe an inverse relationship (the higher the score, the lower the accuracy). For SFDA (Figure 11), we can imagine

Table 3: Average Pearson correlation between accuracy differences and metric score differences across datasets

| Metric | Avg. Pearson $r$ |
|---|---|
| SFDA | 0.430 |
| NLEEP | 0.299 |
| H-Score | 0.101 |
| LogME | 0.047 |
| GBC | $-0.147$ |
| TransRate | $-0.178$ |

Table 4: Original results from LogME NLP Experiments.

| | task | RoBERTa | RoBERTa-D | uncased BERT-D | cased BERT-D | ALBERT-v1 | ALBERT-v2 | ELECTRA-base | ELECTRA-small | $\tau_w$ |
|---|---|---|---|---|---|---|---|---|---|---|
| MNLI | Accuracy | 87.6 | 84.0 | 82.2 | 81.5 | 81.6 | 84.6 | 79.7 | 85.8 | - |
| | LogME | -0.568 | -0.599 | -0.603 | -0.612 | -0.614 | -0.594 | -0.666 | -0.621 | 0.66 |
| QNLI | Accuracy | 92.8 | 90.8 | 89.2 | 88.2 | - | - | - | - | - |
| | LogME | -0.565 | -0.603 | -0.613 | -0.618 | - | - | - | - | 1.00 |
| SST-2 | Accuracy | 94.8 | 92.5 | 91.3 | 90.4 | 90.3 | 92.9 | - | - | - |
| | LogME | -0.312 | -0.330 | -0.331 | -0.353 | -0.525 | -0.447 | - | - | 0.68 |
| CoLA | Accuracy | 63.6 | 59.3 | 51.3 | 47.2 | - | - | - | - | - |
| | LogME | -0.499 | -0.536 | -0.568 | -0.572 | - | - | - | - | 1.00 |
| MRPC | Accuracy | 90.2 | 86.6 | 87.5 | 85.6 | - | - | - | - | - |
| | LogME | -0.573 | -0.586 | -0.605 | -0.604 | - | - | - | - | 0.53 |
| RTE | Accuracy | 78.7 | 67.9 | 59.9 | 60.6 | - | - | - | - | - |
| | LogME | -0.709 | -0.723 | -0.725 | -0.725 | - | - | - | - | 1.00 |

linear mappings that fit to most datasets. This is also reflected by comparatively large correlation between the SFDA score and the accuracy in Figure 4. Transrate's scores are all over the place (cf. Figure 12), and the H-score exhibits some linear relationships for the Cifar-10, Cifar-100 and Food-101 datsets (cf. Figure 13).

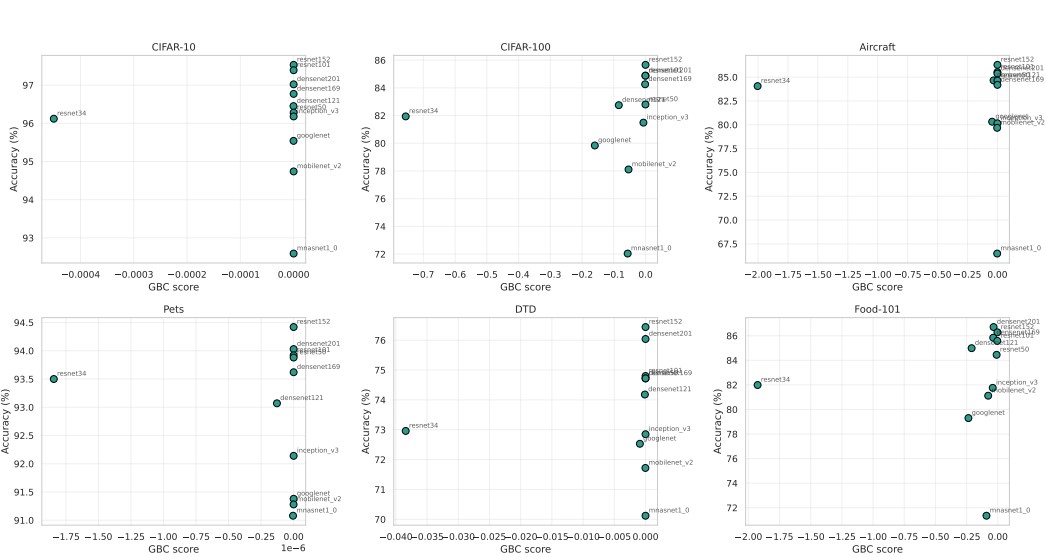

Figure 8: Plot of GBC scores against the ground truth accuracy.

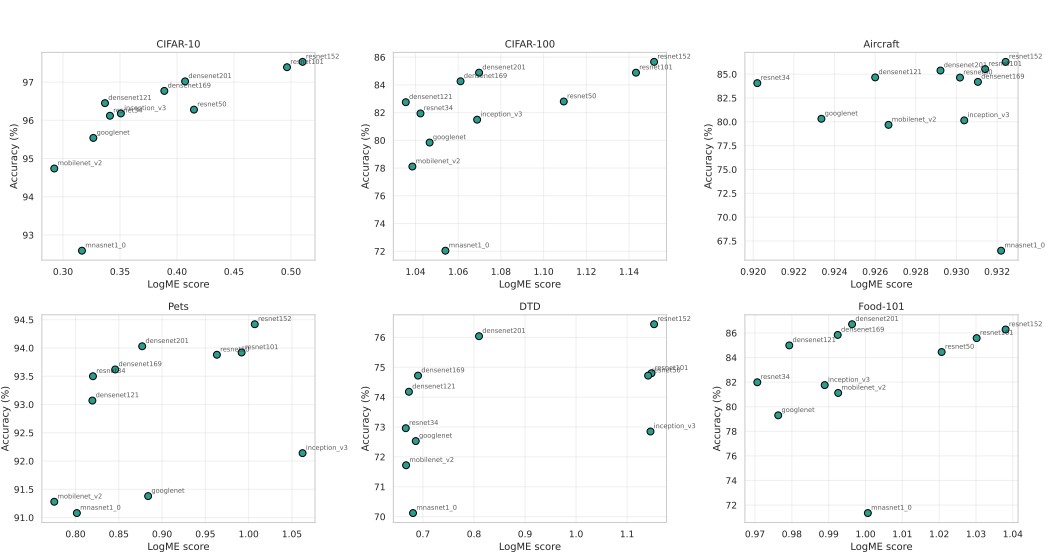

Figure 9: Plot of LogME scores against the ground truth accuracy.

## F   LLM USAGE

LLMs have been used for proofreading, and finding typos.

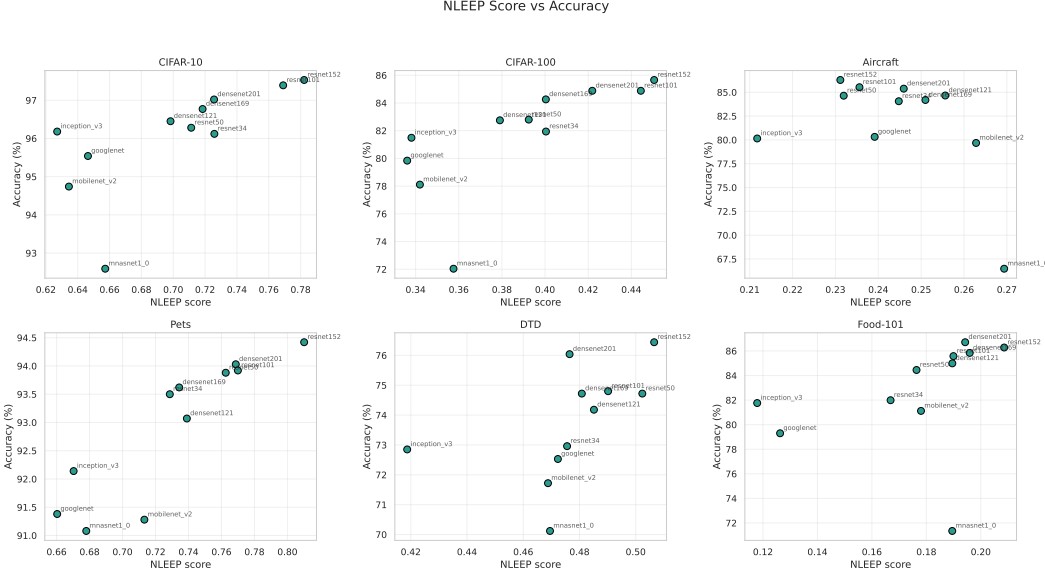

Figure 10: Plot of NLEEP scores against the ground truth accuracy.

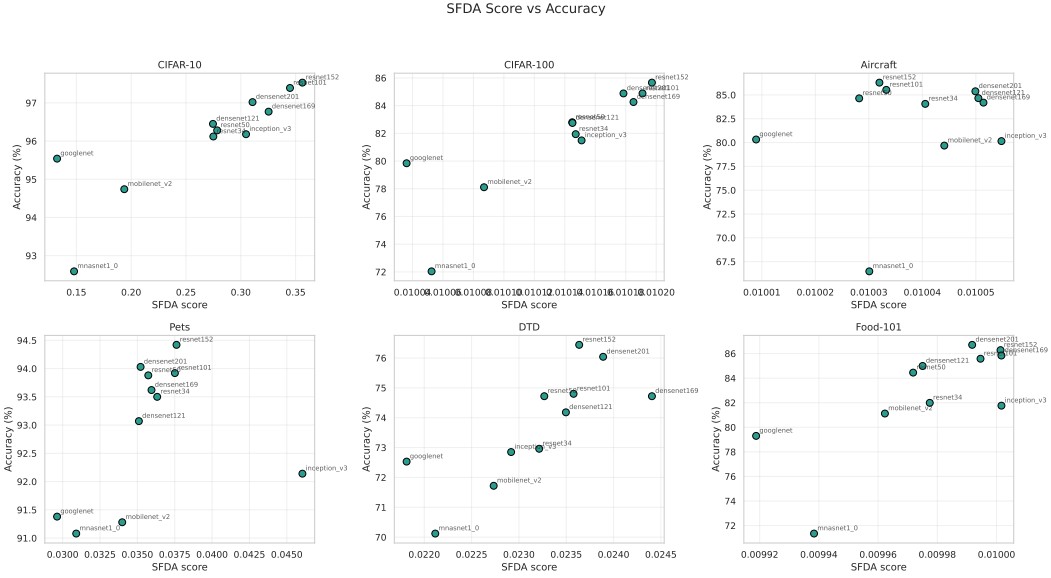

Figure 11: Plot of SFDA scores against the ground truth accuracy.

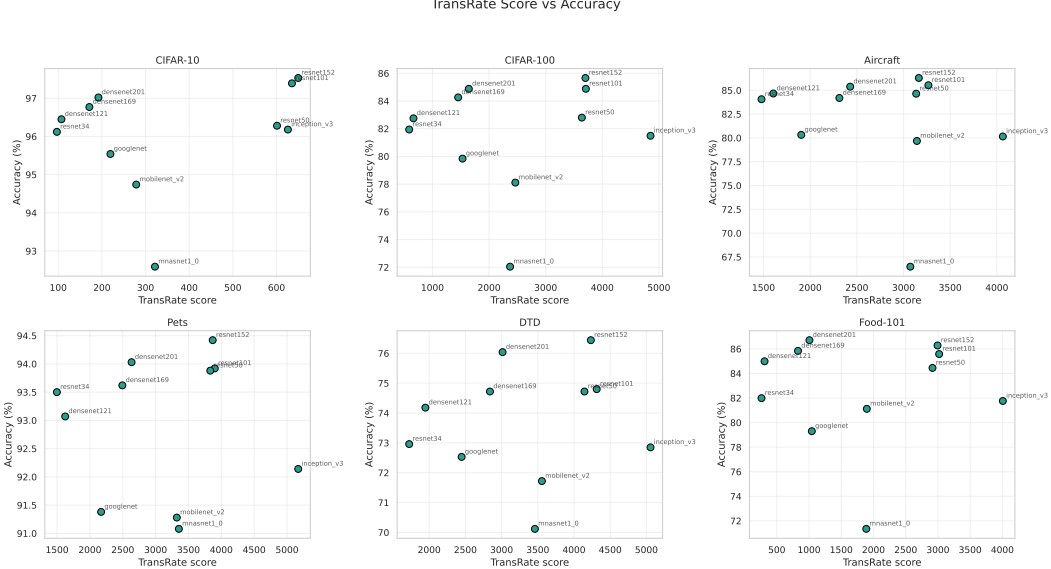

Figure 12: Plot of Transrate scores against the ground truth accuracy.

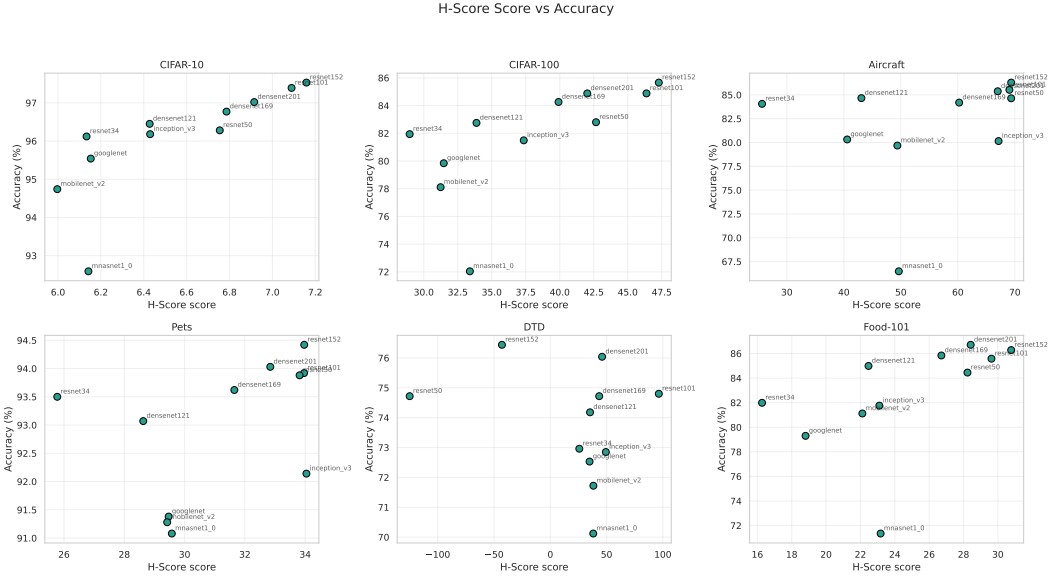

Figure 13: Plot of H-score against the ground truth accuracy.

