# OpenReview forum: "How NOT to benchmark your SITE metric: Beyond Static Leaderboards and Towards Realistic Evaluation."
_ICLR.cc/2026/Conference — ICLR 2026 Poster_

### Official Review · Reviewer_ULr8 · 2025-10-24

**Soundness:** 3
**Presentation:** 2
**Contribution:** 3
**Rating:** 6
**Confidence:** 3

**Summary:**

This paper discusses the evaluation of metrics for Source Independent Transferability Estimation. These metrics attempt to predict pretrained models' downstream performance after finetuning on a given dataset. The paper raises three points of criticism aimed at current evaluation practices based on a small set of models and datasets: First, the set of included models is often unrealistic and dominated by differently sized versions of a small set of architectures. Evaluations results are heavily dependent on the set of included models. Second, a heuristic static model ranking outperforms all evaluated methods. Third, current evaluation metrics focus on rank correlations, partially masking failures to predict the magnitude of performance differences.

**Strengths:**

- The "static ranking" experiment relatively convincingly demonstrates that the current evaluation setup for SITE metrics is flawed

**Weaknesses:**

- The paper could be strenghtened by producing an actual setup that mitigates the discussed issues.
- The writing could be improved in various places.
   - For example, the discussion mentions a bunch of datasets without citations
   - Some claims in the paper are also unsupported (by either citations or arguments). Example: "In addition, SDTE metrics are not reliable when the discrepancy between the source and target datasets is very high"
    - As a minor nitpick, the paper could also benefit some additional proofreading in terms of grammar/sentence structure. For example: "For purpose of this study we examine the following metrics on standard benchmark"

**Questions:**

How exactly was the static ranking chosen? The paper says "based on the model size and an alternation of ResNet and DenseNet model families", but there seem to be two densenets followed by two resnets at ranks 4-7.

---

> ### Author Response · Authors · 2025-11-20
>
> > The paper could be strengthened by producing an actual setup that mitigates the discussed issues.
> In response to your suggestions, we have constructed a new, more robust benchmark to evaluate SITE metrics.
>
> We began by selecting the Meta-Album suite, which contains 30 datasets. To ensure our benchmark effectively differentiates between model performances, we first evaluated a diverse set of models
> (Twins-SVT, XCiT, CoaT, DeiT, MaxViT, and MViTv2) on all 30 datasets. We identified and subsequently removed datasets that were "too easy," meaning multiple models achieved near-perfect accuracy. This ceiling effect is not ideal for assessing the sensitivity of SITE metrics.
> Our final benchmark, curated according to best practices, consists of the 15 more challenging datasets that produced a wide range of accuracy scores. These datasets are: Sports, Plant Village, RESISC, Insects, PanNuke, MPII Human Pose Dataset, Fungi, RSD, Boats, Plant Doc, Stanford Actions, DTD, Subcellular Human Protein Dataset (PRTA), Insects SPIPOLL, and Dogs.
> The results of our experiments on this new benchmark are presented in Table 2. Our findings confirm that no single SITE metric performs consistently well across these datasets. We believe this benchmark, developed in light of your comments, provides a more reliable foundation for evaluating SITE metrics and can serve as a model for creating similar benchmarks in other domains like NLP and information retrieval.
>
> > The writing could be improved in various places.For example, the discussion mentions a bunch of datasets without citations
>
> We have fixed this, thank you
>
> > Some claims in the paper are also unsupported (by either citations or arguments). Example: "In addition, SDTE metrics are not reliable when the discrepancy between the source and target datasets is very high"
>
> We have provided appropriate citation for this claim
> > As a minor nitpick, the paper could also benefit some additional proofreading in terms of grammar/sentence structure. For example: "For purpose of this study we examine the following metrics on standard benchmark"
>
> We have proofread the paper again and have made several edits (grammar edits are not in blue)
>
> > How exactly was the static ranking chosen? The paper says "based on the model size and an alternation of ResNet and DenseNet model families", but there seem to be two densenets followed by two resnets at ranks 4-7.
>
> We thank the reviewer for spotting this point of confusion. The reviewer is correct that the ranking in Figure 3 does not perfectly match the alternating heuristic we described. This is because Figure 3 shows the ground truth ranking of models based on their actual finetuned performance, which varies by dataset.
> Our static ranker, in contrast, uses a single, fixed, data-agnostic ranking for all datasets, which is defined in the text (L164 in the updated manuscript): ResNet-152 ≻ DenseNet-201 ≻ ResNet-101, etc. This heuristic strictly alternates between the ResNet and DenseNet families, ordered by size.
> We realize that placing the description of the static ranker far from the figure caused this confusion. We have now clarified the caption of Figure 3 to explicitly state that it shows the ground truth performance.

---

> ### Comment · Reviewer_ULr8 · 2025-11-21
>
> Thank you for the reply; it seems like most of my points have been addressed.
>
> I maintain my score, as to the best of my knowledge, despite the score descriptions, "6" means accept while "8" means "spotlight/oral", and I find the paper's scope a little to narrow for the latter.

---

> > ### Author Response · Authors · 2025-11-22
> >
> > We're glad to know that we have resolved your concerns. While we would like to add a note regarding the comment on the narrow scope of our work (also mentioned as a reply to Reviewer GpHM):
> >
> > While these scores are also primarily developed for computer vision, multiple works have also used them for different tasks, such as :
> >
> > - An Approach Using MMD and Kernel Methods, CVPR2024, which used LogME and GBC as baselines for IR tasks
> > - Model Recycling Framework for Multi-Source Data-Free Supervised Transfer Learning, Wang et al, MLSP 25, uses LogME as a part of their experimental setup
> > - On the Utility of Existing Fine-Tuned Models on Data-Scarce Domains TMLR 2025, Alam et al uses LEEP scores to validate the models themselves.
> >
> > Many more examples show how the transferability estimation benchmark and how we develop them are important and affect broader ML ecosystems and even interdisciplinary fields like IR and Signal processing. We hope that this information broadens the impact of our contribution.

---

### Official Review · Reviewer_fF5c · 2025-10-30

**Soundness:** 3
**Presentation:** 3
**Contribution:** 2
**Rating:** 4
**Confidence:** 3

**Summary:**

The authors propose a deeper look into the problems in the assumptions made for evaluating transferability estimation. They show how the the status quo benchmark framework (datasets, models, evaluation protocols) is faulty will 3 concrete emperical critiques and make concrete suggestions for good practises that future works can focus on.

**Strengths:**

1. Well-motivated problem statement: shifting from SDTE to SITE is timely and important as we know less and less about the true distribution of training data. Also, with scale, distribution matching will only get more expensive.

2. Probing into the correlation between the difference in magnitude of accuracy and tranferability is a critical analysis the paper did quite well. With more fine-grained studies, it can be quite insightful to practioners.

**Weaknesses:**

My main issues with this paper revolve around the lack of extensive experimentation: to make the strong claim that a form of benchmarking is "fundamentally flawed" requires significantly more analysis than currently provided.

1. Even though the paper prescribes including ViT, ConvNeXt, MLP-Mixer etc, the core experiments to determine the core claim does not extend beyond the standard set of models (ResNet, DenseNet, etc). Though intuitive, we need emperical evidence that the three critiques will hold true with a diverse pool of models.

2. Related works such as [1] have identified similar problems, such as "bias toward high-capacity models" and the unrealistic model search space, in addition to other relevant problems this work has not addressed such as the lack of fully labelled target datasets in real-world scenarios. They propose solutions such as adding self-supervised models and evaluating transferability estimation on model weights before and after optimisation using the Wasserstein Distance(even though their analysis focusses on SDTE, both these solutions are applicable to SITE). Existing literature already extends the scope of this work by providing concrete solutions to fix the evaluation setup, calling the novelty of this work into question.

3. Finetuning needs to be more vigorous: Small changes in seed, optimizer, number of epochs, etc can produce statistically significant variance in model performance. The paper does not but should report independent fine-tuning sweeps and checkpoints and report mean &plusmn; standard deviation to check for this sensitivity. Also, the same seeds should be applied to all models and model families.

4. I disagree with the claim that larger models naturally outperform their smaller counterpart. Recently, [2] has shown that sample-level benchmarking showcases interesting scenarios where older versions of the same model or smaller models of the same family outperform their newer iteration/larger size counterpart for specific subsets of the evaluation datapool. This brings into question whether the static ranking based on model size is robust to more specific data subsets.

5. The NLP results feel like a footnote in the appendix. It should be as rigorous as the image classification experiments. In general, for a paper claiming transferability estimation metrics are flawed and that the insights are applicable to other domains and tasks, it is in the scope of the work to demonstrate that. Related works such as [3] do show analysis on medical image classification.


Minor points:
1. L137: using the same notation n for (T_n) and data (x_n, y_n).

[1] Kazemi et al, Benchmarking Transferability: A Framework for Fair and Robust Evaluation, 2025

[2] Ghosh et al, ONEBench to Test Them All: Sample-Level Benchmarking Over Open-Ended Capabilities, ACL 2025

[3] Chaves et al, Back to the Basics on Predicting Transfer Performance, 2024

**Questions:**

1. How does the fidelity affect top models? Practitioners generally care about the top performing models, so I would be curious to see how $\Delta$Acc - $\Delta$T correlation for model pairs where at least one of the models is in the top-k or the bottom-k. This would tell us if fidelity fails in the current evaluation setup for important models.

2. Re-iterating from W4, does the static leaderboard still outperform SITE metrics for granular sample-level datapoints?

---

> ### Author Response · Authors · 2025-11-20
>
> We thank Reviewer for their detailed feedback and for pushing us to strengthen our empirical claims. The reviewer’s central critique that our paper requires more extensive experimentation to validate its claim is well-taken. In direct response, we have conducted a major new set of experiments by constructing a robust benchmark with the diverse, modern architectures the reviewer suggested. These new results, which are now integrated into the paper, confirm that our critiques hold and significantly strengthen our contribution. We will address this major update first, followed by the other valuable points raised regarding novelty and experimental details
>
> > Even though the paper prescribes including ViT, ConvNeXt, MLP-Mixer etc, the core experiments to determine the core claim does not extend beyond the standard set of models (ResNet, DenseNet, etc). Though intuitive, we need emperical evidence that the three critiques will hold true with a diverse pool of models.
>
>
> In response to your suggestions, we have constructed a new, more robust benchmark to evaluate SITE metrics.
> We began by selecting the Meta-Album suite, which contains 30 datasets. To ensure our benchmark effectively differentiates between model performances, we first evaluated a diverse set of models
> (Twins-SVT, XCiT, CoaT, DeiT, MaxViT, and MViTv2) on all 30 datasets. We identified and subsequently removed datasets that were "too easy," meaning multiple models achieved near-perfect accuracy. This ceiling effect is not ideal for assessing the sensitivity of SITE metrics.
> Our final benchmark, curated according to best practices, consists of the 15 most challenging datasets that produced a wide range of accuracy scores. These datasets are: Sports, Plant Village, RESISC, Insects, PanNuke, MPII Human Pose Dataset, Fungi, RSD, Boats, Plant Doc, Stanford Actions, DTD, Subcellular Human Protein Dataset (PRTA), Insects SPIPOLL, and Dogs.
> The results of our experiments on this new benchmark are presented in Table 2. Our findings confirm that no single SITE metric performs consistently well across these datasets. We believe this benchmark, developed in light of your comments, provides a more reliable foundation for evaluating SITE metrics and can serve as a model for creating similar benchmarks in other domains like NLP and information retrieval.
>
> > Related works such as [1] have identified similar problems, such as "bias toward high-capacity models" and the unrealistic model search space, ....
>
> While Kazemi et al. [1] also identify issues like model complexity, our work is fundamentally different in its scope and contribution.
>
> Diagnosis vs. Observation: While Kazemi et al. briefly mention that the standard benchmark is not complex enough, they do not specify why that is the case. Our primary contribution is to systematically diagnose and empirically quantify criteria that have an impact on SITE benchmarks. We introduce novel analyses, such as the "static ranker" baseline and model ablation studies, to prove that the standard SITE benchmark is effectively "solved" by trivial heuristics. Our work demonstrates why and how these flaws lead to artificially inflated performance, a diagnosis not present in [1].
>
> Critique of the Critique: As the reviewer notes, existing literature identifies problems but often falls into the same traps. We observe that even [1], after critiquing the model space, proceeds to evaluate on a similar, undiversified set of models. This trend leads to using the same experimental setup for future metrics. Our work's explicit goal is to break this cycle by not only critiquing the protocol but also establishing a list of best practices when designing a benchmark for transfer learning.
>
> [1] Kazemi et al, Benchmarking Transferability: A Framework for Fair and Robust Evaluation, 2025

---

> > ### Author Response · Authors · 2025-11-20
> > **Reply Continued**
> >
> > > Finetuning needs to be more vigorous: Small changes in seed, optimizer, number of epochs, etc can produce statistically significant variance in model performance. The paper does not but should report independent fine-tuning sweeps and checkpoints and report mean ± standard deviation to check for this sensitivity. Also, the same seeds should be applied to all models and model families.
> >
> > We agree that reporting results over multiple seeds is a critical component of robust evaluation. However, for the specific goal of this paper, critiquing the standard benchmark as it is currently used, it was a deliberate methodological choice to replicate the exact fine-tuning protocol from prior work (e.g., LogME, TransRate, and [1]). In this setup, the same seeds were applied to all models and model families.
> >
> > Our aim is to show that the benchmark is flawed even under its own idealized conditions. Using a more robust finetuning protocol would deviate from the standard setup and allow defenders of the current benchmark to argue that our critique is not an "apples-to-apples" comparison. By using the established protocol, we demonstrate that the benchmark's flaws are inherent to its design, independent of finetuning variance.
> >
> > That said, we fully agree with the reviewer's sentiment. In our "Best Practices" section, we also emphasize on finetuning and also identify that current site metrics take no account of finetuning in their computation, like co tuning, full fine tuning etc.
> >
> > [1] Kazemi et al, Benchmarking Transferability: A Framework for Fair and Robust Evaluation, 2025
> >
> > > I disagree with the claim that larger models naturally outperform their smaller counterpart. Recently, [2] has shown that sample-level benchmarking showcases interesting scenarios where older versions of the same model or smaller models of the same family outperform their newer iteration/larger size counterpart for specific subsets of the evaluation datapool. This brings into question whether the static ranking based on model size is robust to more specific data subsets.
> >
> > We thank the reviewer for pointing this out, we have rewritten our critique in section 5.1 and avoid the overstatement that larger models naturally outperform their smaller counterparts more accurately to our specific setup. We thank the reviewer again for pointing that out, so we avoid the overstatement.
> >
> > > The NLP results feel like a footnote in the appendix. It should be as rigorous as the image classification experiments. In general, for a paper claiming transferability estimation metrics are flawed and that the insights are applicable to other domains and tasks, it is in the scope of the work to demonstrate that. Related works such as [3] do show analysis on medical image classification.
> >
> > The reason for the NLP results being in the appendix is that only LogME in the original works has claimed to work for NLP, and other SITE metrics have no such claims, so it will make evaluation of those models for NLP unfair, as the source work never claims to work for NLP. While the purpose of NLP results is to show that the LogME source paper also did not incorporate the right elements for NLP work as well similar to the vision benchmark.
> >
> > > L137: using the same notation n for (T_n) and data (x_n, y_n).
> >
> > Thank you, we have fixed it in the manuscript now.

---

> > > ### Author Response · Authors · 2025-11-20
> > > **Questions:**
> > >
> > > > How does the fidelity affect top models? Practitioners generally care about the top performing models, so I would be curious to see how Acc - T correlation for model pairs where at least one of the models is in the top-k or the bottom-k. This would tell us if fidelity fails in the current evaluation setup for important models.
> > >
> > > We thank the reviewer for this suggestion. We have performed the requested fidelity analysis on pairs of models where models are in the top-4 and bottom-4 ranks of the standard benchmark. Our findings further strengthen the paper's claims:
> > > For top-k models: We found that the correlation between a SITE score gap (ΔScore) and the actual accuracy gap (ΔAccuracy) is highly inconsistent and often weak. This confirms that even for the most important models, the metrics fail to provide a meaningful sense of how much better one model is than another.
> > > For bottom-k models: The correlation was generally even weaker, with the performance of all metrics declining significantly.
> > > These results are critical because they show that fidelity fails precisely where it matters most: when a practitioner is deciding between top-performing models. The inability of SITE scores to reliably predict the magnitude of performance gains makes them impractical for real-world cost-benefit analysis. We have added this analysis to the appendix and summarized the findings in Appendix C.1.
> > >
> > >
> > >
> > > > Re-iterating from W4, does the static leaderboard still outperform SITE metrics for granular sample-level datapoints?
> > >
> > >  Our current work did not extend to a sample-level analysis as performed by Ghosh et al. [2]. Our primary goal was to demonstrate that the current, standard evaluation protocol, which operates at the full dataset level, is so flawed that a completely data-agnostic static ranker can outperform sophisticated SITE metrics. If a trivial heuristic works at the macro-level, the benchmark itself is not challenging enough.
> > > [2] Ghosh et al, ONEBench to Test Them All: Sample-Level Benchmarking Over Open-Ended Capabilities, ACL 2025

---

> > > > ### Author Response · Authors · 2025-11-28
> > > >
> > > > Dear reviewer,
> > > >
> > > > Just a reminder to respond to our rebuttal. We believe to have addressed most of your concerns.

---

### Official Review · Reviewer_GpHM · 2025-10-31

**Soundness:** 4
**Presentation:** 4
**Contribution:** 4
**Rating:** 8
**Confidence:** 3

**Summary:**

This paper provides an empirically-backed critique of existing evaluations for SITE (source-independent transferability estimation). This problem concerns how to estimate which of a pool of models will perform best on a target domain after finetuning. The paper makes several well-received points: existing evaluations tend to use overly simple pools of models and datasets. As a result, naive baselines like a hand-crafted static ranking of models can outperform highly sophisticated SITE methods. Popular ranking metrics also tend to understate how poorly SITE scores reflect actual differences in finetuned model accuracies (on an absolute scale, not just in terms of rank ordering). The paper offers several constructive recommendations for how to improve evaluation in the space. I believe the strongest point is that evaluations must capture a meaningful degree of variance in finetuned model performance across different domains, i.e. rank orderings of finetuned models on target datasets should show a meaningful degree of variance that SITE metrics should then explain.

**Strengths:**

- Very important: The paper is well written and very clear.
- Very important: The paper’s experiments get straight to the point and effectively undermine the validity an existing, popular approach to SITE benchmarking.
- Very important: The recommendations are fairly actionable and should push work on this problem in the right direction.

**Weaknesses:**

- Of some importance: I am not deeply familiar with work on SITE and so a small concern of mine is that this paper is strawmanning what is a small and not overly influential line of research. Yes, I agree that these papers are being published in top venues. But is this really the mainstream way that people think about capturing low-dimensional properties of AI’s adaptability to downstream tasks? Forgive me for comparing this problem area to LLM evaluations, but I think what people tend to put stock in these days is a small set of benchmark scores for an AI system. A set of 3-4 benchmark scores is taken as gospel for which model is most “intelligent”, and therefore going to perform best when tailored to a specific downstream task. Sure, there is clever ML work going on to understand properties of vision model representations and how they relate to downstream performance. But this is not the main thing I have in mind when I think of how people assess broad transferability of AI systems.
- Of minor importance: I had a small quibble with this point: “practitioners are not interested in knowing whether they should use a ”bigger vs. smaller” ResNet, but which architecture performs best under a fixed size budget.” I think this point is stated a little more elegantly in the recommendations section: “To isolate architectural inductive bias, models should be matched on computational budgets (e.g., parameter counts or FLOPs).” In reality, practitioners care about a lot of things, including size, training speed, inference speed, ease of access to models, interoperability with common training and deployment libraries, etc. I think the claim in the recommendations section helps point out what kind of model pool would lend support to scientific conclusions about predicting downstream adaptability of different model architectures. Still more consideration might go into constructing a model pool that reflected all the various axes of variance that practitioners care about (which is more than just performance vs. size).

**Questions:**

- please feel free to comment on the noted minor weaknesses/concerns above
- typo: as two scorings with induce the same order can differ in their evaluation merely due to calibration

---

> ### Author Response · Authors · 2025-11-20
> **Reply to Reviewer GpHM**
>
> We thank Reviewer GpHM for the supportive and constructive review. We are delighted that the reviewer found our critiques effective and our recommendations actionable. Your primary concern regarding the influence of SITE research is an important one, and we appreciate the opportunity to elaborate on the field's growing relevance and, consequently, the timeliness of our contribution.
>
> > Of some importance: I am not deeply familiar with work on SITE and so a small concern of mine is that this paper is strawmanning what is a small and not overly influential line of research. Yes, I agree that these papers are being published in top venues. But is this really the mainstream way that people think about capturing low-dimensional properties of AI’s adaptability to downstream tasks? Forgive me for comparing this problem area to LLM evaluations, but I think what people tend to put stock in these days is a small set of benchmark scores for an AI system. A set of 3-4 benchmark scores is taken as gospel for which model is most “intelligent”, and therefore going to perform best when tailored to a specific downstream task. Sure, there is clever ML work going on to understand properties of vision model representations and how they relate to downstream performance. But this is not the main thing I have in mind when I think of how people assess broad transferability of AI systems.
> The reviewer's concern about whether this is a "small... line of research" is valid, and we appreciate the opportunity to clarify its high-leverage and growing importance.
>
> On the "Strawman" Concern and the Field's Relevance: Our work is not critiquing a small, isolated field. On the contrary, it is an active and rapidly expanding area of research, with new methods appearing in top-tier venues. As the reviewer notes, these papers are being published in ICML, NeurIPS, and CVPR. This is not slowing down; in just the last few months, new methods have been proposed:
>
> 1. Feature Space Perturbation: A Panacea to Enhanced Transferability Estimation, WACV 2025
> 2. A High-Dimensional Statistical Method for Optimizing Transfer Quantities in Multi-Source Transfer Learning NeurIPS 2025
> 3. Model Transferability Informed by Embedding’s Topology NeurReps 2025
> 4. Unified Transferability Metrics for Time Series Foundation Models NeurIPS 2025
> 5. PEFTDiff: Diffusion-Guided Transferability Estimation for Parameter-Efficient Fine-Tuning ICCV 2025
>
> Critically, these new works show the problem expanding beyond classic CV to time series, diffusion models, and parameter-efficient fine-tuning (PEFT). This brings us to our paper's central, urgent point: these new 2025 papers are still being validated using the very same flawed "standard benchmark" protocols that our paper critiques.
>
> Our work is therefore not attacking a strawman but attempting to correct the trajectory of a rapidly growing field before these flawed evaluation practices become further entrenched and metastasize into these new domains.
>
> While these scores are also primarily developed for computer vision, multiple works have also used them for different tasks like
> - An Approach Using MMD and Kernel Methods, CVPR2024, which used LogME and GBC as baselines for IR tasks
> - Model Recycling Framework for Multi-Source Data-Free Supervised Transfer Learning, Wang et al, MLSP 25, uses LogME as a part of their experimental setup
> - On the Utility of Existing Fine-Tuned Models on Data-Scarce Domains TMLR 2025, Alam et al uses LEEP scores to validate the models themselves.
>
> There are many more examples that show how the transferability estimation benchmark and how we develop them are important and affect broader ML ecosystems and even interdisciplinary fields like IR and Signal processing.
> Indeed, SITE metrics have been seen as the gospel for model selection, and we are questioning this and showing that a simple heuristic has high performance on these metrics.
>
> >  Of minor importance: I had a small quibble with this point: “practitioners are not interested in knowing whether they should use a ”bigger vs. smaller” ResNet ....
>
> Our core argument is that to isolate architectural advantages, a key scientific goal of SITE is that models should be compared on a level playing field (e.g., matched for FLOPs or parameter counts). We have revised the text in the "Best Practices" section to reflect this more nuanced perspective, emphasizing the need for benchmarks that consider these multifaceted, real-world constraints. For our suggested benchmark, we have incorporated the models according to the parameter size, as training and inference speed are hard to obtain without training the models at first, whereas in the SITE setting, that information is generally not available.

---

> > ### Comment · Reviewer_GpHM · 2025-11-23
> >
> > Thanks for the response! I have increased my confidence rating from 3 to 4.

---

### Official Review · Reviewer_uC4E · 2025-11-03

**Soundness:** 4
**Presentation:** 2
**Contribution:** 4
**Rating:** 6
**Confidence:** 2

**Summary:**

This paper argues that the standard evaluation protocol for Source-Independent Transferability Estimation (SITE) is misspecified and overstates progress. Unlike Agostinelli et al., this work targets the protocol rather than metric design.

The authors show three failures:

- **1. Unrealistic model space.** Common SITE pools are dominated by scaled variants of a few CNN families (chiefly ResNet and DenseNet). Removing the largest models sharply drops Kendall’s τ_w across methods, exposing brittleness and a “bigger-is-better” effect that drives rankings.

- **2. Static leaderboard.** A trivial, dataset-agnostic ranking -- argely by model size and family -- matches or beats SITE metrics across datasets. The same few large models (e.g., ResNet-152) repeatedly top the charts; a fixed size-aware ranker attains higher τ_w than all evaluated SITE metrics.

- **3. No fidelity checks.** SITE scores are not tested for decision fidelity -- whether score gaps map to meaningful accuracy gains. Pairwise ∆score correlates weakly with pairwise ∆accuracy, making score differences hard to interpret for cost–benefit trade-offs.

The paper ends with best-practice recommendations and a checklist for better evaluation.

**Strengths:**

- **Timely and important.** Evaluation design choices meaningfully shape this subfield; focusing on protocol is valuable in its own right.

All the three findings seem correct and fatal:
- Ablations show over-represented large models inflate results (Fig. 2).
- The static-ranker baseline dominates (Table 1) and should be fairly easy to reproduce.
- The ∆score vs ∆accuracy study addresses an often-ignored need and shows that score gaps frequently fail to predict accuracy gains (Fig. 4; App. B).

It's correct and likely impactful on the field of source-independent transferability estimation.

**Weaknesses:**

**[Critical] Could the paper implement the fixes recommended instead of best practices?**

The critiques are actionable, yet the paper stops short of fixing it. Best-practices section seems unsuitable as it reads as advice masquerading as work that is not done.

- If the pool over-indexes ResNet/DenseNet, please propose and release a diversified model space (include ViTs and other modern families). Alternatively consider a two-stage aggregation: Compute SITE within each architecture family, then aggregate fairly across families. But a new benchmark, if proposed, instead of simply this critique could be widely adapted.

- If a static ranker “solves” current datasets, expand the dataset suite (e.g., include harder, real-world transfer targets such as those discussed in *Does Progress on ImageNet Transfer to Real-World Datasets?* by Fang et al.). Leaderboard suites from *Benchmark suites instead of leaderboards for evaluating AI fairness* by Wang et al. might also be valuable.

- For fidelity, propose an actionable fidelity metric grounded in decision quality (∆score vs ∆accuracy study already seemed promising). To provide some straws in the wind -- social-choice theory offers rank-aggregation tools with known axioms and trade-offs; some classical criteria may fit SITE’s needs.

If fixed, the resulting benchmark could be potentially quite valueable for the next generation of SITE algorithms.

**[Major] Central findings #1 and #2 already seems well-known.**

- The strong performance of dataset-agnostic rankings (by size or ImageNet top-1) echoes established findings -- e.g., *Do Better ImageNet Models Transfer Better?* (Kornblith et al., CVPR’19) and *Do ImageNet Classifiers Generalize to ImageNet?* (Recht et al., ICML’19). Could the work clarify what is new beyond confirming known transfer trends.
- Lots of similar work: *Back to the Basics on Predicting Transfer Performance*, *Scalable Diverse Model Selection for Accessible Transfer Learning* (relevant to finding #1) are not cited or differentiated from. Could the authors detail the core differences to past work?

**[Minor] Reporting and metrics.**

- How large are the uncertainty bands (confidence intervals and significance tests) for τ_w to indicate whether observed differences are meaningful.
- The work seemed to need more polishing to concisely state the message. Please confirm whether my understanding and critiques are correct, I had to read this multiple times to understand.

**Questions:**

Please address weakness 1 and 2.

Overall, a well-argued, well-executed critique of current SITE evaluation with compelling evidence -- solid work in my opinion. The work would be much stronger if it shipped a concrete repaired benchmark (diversified model pool, harder datasets, and a fidelity metric), rather than "best-practices" guidance.

---

> ### Author Response · Authors · 2025-11-20
> **Reply to reviewer uC4E (Benchmark and weaknesses)**
>
> "We thank Reviewer uC4E for their detailed and insightful review. We are grateful for the reviewer's excellent summary of our work and for recognizing our findings as 'correct and fatal.' The reviewer's most critical point is that our work would be much stronger if we moved beyond critique and implemented the fixes we recommend. We wholeheartedly agree. In direct response, we have made a major addition to our paper: we have constructed and evaluated a new, robust benchmark that follows our proposed best practices. This new contribution directly addresses the primary weakness identified, and we will detail it first before addressing the reviewer's other important points on novelty and related work.
>
> > Could the paper implement the fixes recommended instead of best practices?
>
>
> The critiques are actionable, yet the paper stops short of fixing it. Best-practices section seems unsuitable as it reads as advice masquerading as work that is not done. If the pool over-indexes ResNet/DenseNet, please propose and release a diversified model space (include ViTs and other modern families). Alternatively consider a two-stage aggregation: Compute SITE within each architecture family, then aggregate fairly across families. But a new benchmark, if proposed, instead of simply this critique could be widely adapted.
> Reply
> In response to your suggestions, we have constructed a new, more robust benchmark to evaluate SITE metrics.
> We began by selecting the Meta-Album suite, which contains 30 datasets. To ensure our benchmark effectively differentiates between model performances, we first evaluated a diverse set of models
> (Twins-SVT, XCiT, CoaT, DeiT, MaxViT, and MViTv2) on all 30 datasets. We identified and subsequently removed datasets that were "too easy," meaning multiple models achieved near-perfect accuracy. This ceiling effect is not ideal for assessing the sensitivity of SITE metrics.
> Our final benchmark, curated according to best practices, consists of the 15 more challenging datasets that produced a wide range of accuracy scores. These datasets are: Sports, Plant Village, RESISC, Insects, PanNuke, MPII Human Pose Dataset, Fungi, RSD, Boats, Plant Doc, Stanford Actions, DTD, Subcellular Human Protein Dataset (PRTA), Insects SPIPOLL, and Dogs.
> The results of our experiments on this new benchmark are presented in Table 2. Our findings confirm that no single SITE metric performs consistently well across these datasets. We believe this benchmark, developed in light of your comments, provides a more reliable foundation for evaluating SITE metrics and can serve as a model for creating similar benchmarks in other domains like NLP and information retrieval.
>
> > If a static ranker “solves” current datasets, expand the dataset suite (e.g., include harder, real-world transfer targets such as those discussed in Does Progress on ImageNet Transfer to Real-World Datasets? by Fang et al.). Leaderboard suites from Benchmark suites instead of leaderboards for evaluating AI fairness by Wang et al. might also be valuable.
>
> We believe as well that this is the correct approach, and our new benchmark contains datasets that are also discussed in Fang et al. The combination of a new model search space and datasets yields a low-performing static ranker: the best static ranker of our new benchmark is low on average (Kendall tau of 0.3).
>
> > For fidelity, propose an actionable fidelity metric grounded in decision quality (∆score vs ∆accuracy study already seemed promising). To provide some straws in the wind -- social-choice theory offers rank-aggregation tools with known axioms and trade-offs; some classical criteria may fit SITE’s needs.
> If fixed, the resulting benchmark could be potentially quite valuable for the next generation of SITE algorithms.
>
> We provide an aggregated person metric as an example, which one can use as a single metric for examining average fidelity across the experiments. This has been added in Appendix C.2. We also added a connection to SCT as a point in future work.

---

> > ### Author Response · Authors · 2025-11-20
> >
> > > The strong performance of dataset-agnostic rankings (by size or ImageNet top-1) echoes established findings -- e.g., Do Better ImageNet Models Transfer Better? (Kornblith et al., CVPR’19) and Do ImageNet Classifiers Generalize to ImageNet? (Recht et al., ICML’19). Could the work clarify what is new beyond confirming known transfer trends.
> >
> > While prior works like Kornblith et al. (2019) and Recht et al. (2019) have indeed shown that better ImageNet models often transfer better, as pointed out in Fang et al., they don’t translate to datasets not scraped from the internet. In our proposed benchmark, such static transfer trends can not be observed. We have also added more information in the Discussion
> > The success of our simple, dataset-agnostic static ranker is not meant to be a new discovery about transfer learning, but rather an empirical proof that the benchmark itself is flawed and fails to pose a meaningful challenge. Our work is the first to systematically demonstrate this failure within the SITE evaluation context and quantify its inflationary effect on reported metric performance. We have clarified this distinction in the updated manuscript.
> >
> >
> > > Lots of similar work: Back to the Basics on Predicting Transfer Performance, Scalable Diverse Model Selection for Accessible Transfer Learning (relevant to finding #1) are not cited or differentiated from. Could the authors detail the core differences to past work?
> >
> > Thank you for pointing us to both these relevant works, we now describe how our work is different from the aforementioned works.
> > Unlike Chaves et al. (2024), who use the same standard benchmark, our work empirically demonstrates the inflationary effect of static leaderboards and provides a diversified benchmark (see Section 8.).
> > Boyla et al. propose a benchmark to identify the best source dataset for a given target dataset in response to their observation that the standard benchmark is too easy. Their benchmark still uses similar models and datasets as the standard benchmark on a much smaller model space(Resnet, googlenet, mnasnet).
> >
> > We have added appropriate references to these works in our related work section.
> >
> > > How large are the uncertainty bands (confidence intervals and significance tests) for τ_w to indicate whether observed differences are meaningful.
> >
> > Is the reviewer referring to confidence intervals for ranking? In that scenario, we can provide the Friedman test.
> >
> > > The work seemed to need more polishing to concisely state the message. Please confirm whether my understanding and critiques are correct. I had to read this multiple times to understand.
> >
> > We appreciate the reviewer's diligence. We have thoroughly proofread and revised the manuscript to improve clarity and conciseness, ensuring our core message is communicated more effectively.
> >
> >
> > Questions:
> >
> > > Please address weakness 1 and 2.
> >
> > We hope to have addressed the weakness in this reply
> >
> > > Overall, a well-argued, well-executed critique of current SITE evaluation with compelling evidence -- solid work in my opinion. The work would be much stronger if it shipped a concrete repaired benchmark (diversified model pool, harder datasets, and a fidelity metric), rather than "best-practices" guidance.
> >
> > Thank you for your kind words, We have incorporated a benchmark with scores for SITE methods and show that current SITE metrics cannot consistently perform well over a well-designed benchmark following best practices.

---

> ### Comment · Reviewer_uC4E · 2025-11-20
> **Addressed my concerns.**
>
> > Our final benchmark, curated according to best practices, consists of the 15 more challenging datasets that produced a wide range of accuracy scores.
>
> > Our findings confirm that no single SITE metric performs consistently well across these datasets. We believe this benchmark, developed in light of your comments, provides a more reliable foundation for evaluating SITE metrics and can serve as a model for creating similar benchmarks in other domains like NLP and information retrieval.
>
> Thanks for incorporating this! This addresses my primary concern. I increase my score.
>
> Hope my review was helpful despite me not working in this area.
>
> P.S.
>
> > Is the reviewer referring to confidence intervals for ranking? In that scenario, we can provide the Friedman test.
>
> Yep, that would work! Thanks.

---

### Author Response · Authors · 2025-11-28
**Message for New AC**

Dear Area Chair,
Thanks for rolling back the scores to keep the process fair.

Our older score was 6,6,8,4, and after discussion, it was 6,8,8,4 (the reviewer with 4 did not engage in discussion yet.) This can be seen in comments as well

- We had two positive score updates—Reviewer uC4E **6→8** and Reviewer **GpHM 3→4(Confidence)** in confidence, this happened before the bug became a widely publicized issue. We didn’t know the bug existed and simply answered their questions.
- Reviewer ULr8 acknowledged that their issues have been resolved and will maintain their score with this statement:
"I maintain my score, as to the best of my knowledge, despite the score descriptions, "6" means accept while "8" means "spotlight/oral."
- Reviewer fF5c did not engage in the discussion, but we believe that we have addressed their issues, as can be seen in the rebuttal.

We would be grateful if you could take the discussion's evolution into account. The upward revision of the scores came only after a constructive dialogue, and we feel these later scores (on top of already positive scores ) show that our paper addresses an important issue, is novel and timely, and to the standard of ICLR.

We would also like to note major changes made in the paper according to reviewers' suggestions:
- Introducing a benchmark that follows best practices.
- Typos fixed

We believe that our work is timely and important for the transfer learning community, and we believe that it has been improved with the ICLR peer review process.

Thank you for your valuable time.

---

### Meta-Review · Area_Chair_XmHR · 2025-12-19

**Summary:**

All reviewers had an initial borderline to positive impression of the paper, where the main strengths were pointed out as the timely nature and importance of the topic, a convincing set of experiments being shown, and the paper being well written. A subset of the reviewers had a shared major concern that the paper identifies challenges, but seems to only provide (actionable) recommendations rather than implementing and testing them directly. A related criticism was that some aspects were thus described (e.g. also in terms of models and datasets), but not included in any empirical analysis. The rebuttal (see comments below) seems to have largely resolved these concerns on top of already positive initial reviews. The AC agrees that this is a meaningfully done and well-written paper that should be accepted.

**Reviewer Concerns:**

Among many more subtle points, reviewers seem to have had three different key concerns: 1. Actionable recommendations vs. actual action, 2. a broader empirical analysis, 3. insufficient descriptions/discussions and lightly limited scope of the paper. All three of these aspects seem to have been addressed, even if to different extent. In particular, the paper has added a section called "benchmark based on best practices", where they actually go beyond recommendations and further conduct a more extensive empirical analysis on 15 datasets. In addition, the discussion was mildly extended, in line with page limitations, but also providing additional clarifications on e.g. static ranking in the appendix. As such, the only "outstanding" point would be one reviewer having a preference towards a broader scope, although this reviewer recommends acceptance nonetheless

**Reviewer Scores:**

Initial scores for this paper were borderline to positive. Two of the reviewers had a chance to engage in discussion before the freeze occurred. One of them raised their rating (prior to the incident and very unlikely as a consequence of it alone) due to the additional content in the rebuttal of the "benchmark based on best practices" section. This reviewer had raised their score from 6->8 prior to the re-roll, leading to two reviews giving a strong acceptance recommendation. The third reviewer gave an initial rating of 6 and was technically satisfied with the rebuttal, but chose to retain the rating because they believed a paper with a score of 8 to be more likely to receive an "oral presentation", rather than a poster (which the reviewer believes the paper should receive due to its scope). Overall, three reviewers are thus very positive towards the paper's acceptance. A final reviewer gave an initial borderline score of 4 and did not have the chance to engage in the discussion prior to the freeze. However, one of their points of criticism revolved around the experimentation, which was substantially extended in the revision. As such, the AC believes that even if the other concerns would have been left unaddressed, the four reviewers would have easily converged to a unanimous recommendation towards acceptance.

---

### Decision · Program_Chairs · 2026-01-26

Accept (Poster)